

# Inter-colony and inter-annual behavioural plasticity in the foraging strategies of a fjord-dwelling penguin—good news in the face of environmental change?

Myrene Otis[1], Thomas Mattern[1,2,3], Ursula Ellenberg[2,3,4], Robin Long[5], Pablo Garcia Borboroglu[3], Philip J. Seddon[1] and Yolanda van Heezik[1]

[1] Department of Zoology, University of Otago, Dunedin, New Zealand
[2] The Tawaki Trust, Dunedin, New Zealand
[3] Global Penguin Society, Puerto Madryn, Argentina
[4] Department of Marine Science, University of Otago, Dunedin, New Zealand
[5] West Coast Penguin Trust, Hokitika, New Zealand

Corresponding authors
Myrene Otis, otismyrene@gmail.com
Thomas Mattern,
t.mattern@eudyptes.net

## ABSTRACT

The Fiordland penguin or tawaki (*Eudyptes pachyrhynchus*) breeds in the complex fjord systems of New Zealand/Aotearoa's southwest, with penguin colonies distributed from fjord entrances to fjord heads, up to 40 km from the ocean. Until recently, little was known about the marine ecology of fjord-breeding tawaki and how access to the fjord environment may impact the species' foraging strategies. We conducted a comparative study of foraging behaviour in chick-rearing tawaki from colonies located at the entrance and further inside Piopiotahi/Milford Sound, one of New Zealand's 14 fjords. Through the attachment of GPS/dive data loggers, dive parameters were examined to determine behavioural differences between the inner fjord colony (Harrison Cove) and the outer fjord colony (Moraine) during 2019 and 2020. Although situated only eight km from each other, the two colonies showed markedly different foraging preferences, with Moraine birds almost exclusively foraging outside the fjord in both years, while Harrison Cove birds primarily foraged within the fjord in 2020 but not in 2019. Tawaki from each colony also displayed contrasting dive behaviour across years, either adopting a strategy of deeper dives with fast velocities, (Harrison Cove in 2019, Moraine in 2020) or shallower dives with slower velocities (Moraine in 2019, Harrison Cove in 2020). Foraging activity and efficiency for both colonies appeared to be greater in 2020 than 2019, although birds foraged differently to achieve this: Harrison Cove birds dived primarily to depths of 0–20 m whereas Moraine birds switched between shallow dives, and deeper dives to 60–120 m of the water column. Notably different environmental conditions in both the ocean and fjord in 2019 *versus* 2020 may have contributed to the behavioural differences across years. Although replication across multiple fjords is necessary in future, these findings highlight that tawaki possess considerable plasticity in their foraging behaviours which could be advantageous for their future survival in a changing climate.

## INTRODUCTION

As climate change intensifies, warming ocean temperatures and perturbations to the marine biome are becoming increasingly prevalent. Climate change is also positively associated with an increase in the severity and frequency of extreme weather events such as the El Niño-Southern Oscillation (ENSO) (*Cai et al., 2015*; *Chiswell & Sutton, 2020*). A consequence of these changes are shifts in prey availability which can offset marine food webs and alter the foraging behaviours of marine predators. Seabirds are especially affected by alterations to the marine environment during the breeding season. Many become central place foragers when they need to frequent land-based nesting sites to provide for their chicks, forcing a reduction in their foraging ranges (*Williams et al., 1992*; *Hull, Hindell & Michael, 1997*; *Trathan et al., 1998*). Previous work on penguins suggests that individuals rely on prey that are readily available close to their colony sites (*Hennicke & Culik, 2005*; *McCutcheon et al., 2011*). Although penguins may extend their foraging ranges considerably during incubation and post-guard, they tend to make daily trips and maintain smaller foraging ranges during guard stage when chicks are most vulnerable and have lower energy demands (*Collins, Cullen & Dann, 1999*; *Poupart et al., 2019*). When range is restricted, adjusting dive performance can be critical to enhancing a bird's foraging success in areas of low prey availability or in response to environmental changes (*Lescroël & Bost, 2005*; *Berlincourt & Arnould, 2015*). To develop conservation strategies for seabirds in the face of significant changes to their ecosystem, it is necessary to acquire a strong knowledge base of their habitat use and the degree of plasticity in their foraging behaviours.

Environmental conditions, and prey availability can vary markedly between different colony locations (*Chiaradia et al., 2007*; *Hoskins et al., 2008*; *Gulka et al., 2020*). A study of rockhopper penguins at three different sites with varying ecological conditions demonstrated the impact that distinct environments can have on daily foraging trip characteristics (*Tremblay & Cherel, 2003*). Often, foraging behaviour of individuals from colonies located relatively close to each other may still diverge significantly (*Chiaradia et al., 2012*; *Lee et al., 2021*). Some researchers theorise that central-place foraging can lead to strong intra-specific competition, which may result in between-colony spatial segregation and differences in foraging behaviour (*Bolton et al., 2019*; *Ito et al., 2021*). One study of little penguins (*Eudyptula minor*) revealed that even individuals from colonies with overlapping foraging ranges showed differences in diet composition, consuming prey that were significantly different in their trophic position and size (*Chiaradia et al., 2012*). It is also possible that individuals from different colonies show plasticity in their dive behaviours, while maintaining similar levels of foraging effort within a season (*Hoskins et al., 2008*). If individuals from one colony are disadvantaged by their site locality, then this will directly impact their reproductive output and fledging success (*Chiaradia et al., 2007*).

The Fiordland penguin (*Eudyptes pachyrhynchus*) or tawaki (Te Reo Māori), is distributed throughout New Zealand's southwestern coast and southern islands (*Mattern & Wilson, 2018*). A recent study highlighted the foraging behaviour of tawaki during the breeding season from an island off the coast of South Westland (*Poupart et al., 2019*) (Fig. 1). Others have compared the foraging behaviours of one colony in the fjord of Milford

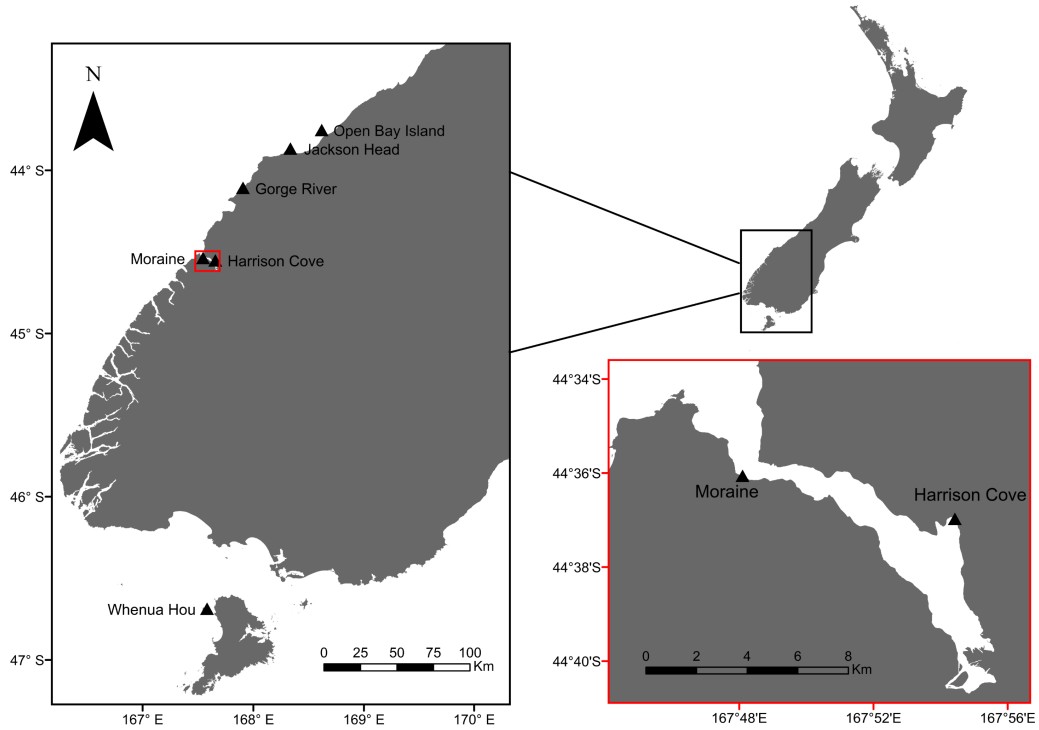

**Figure 1 Map of tawaki breeding colonies that have been studied in the past and those that are part of the current study (Harrison Cove and Moraine).** Red inset map shows a close-up of Milford Sound, with the outer-fiord colony of Moraine and inner-fiord colony of Harrison Cove. Base map data courtesy of LINZ.

Sound to tawaki breeding at sites in South Westland (Jackson Head, Gorge River) and on the southern island of Whenua Hou (*Mattern & Ellenberg, 2016*; *Mattern & Ellenberg, 2017*; *Mattern & Ellenberg, 2018*; Fig. 1). However, little is known about the marine ecology and inter-colony dynamics of tawaki breeding within fjords, even though fjord systems cover a significant proportion of their breeding range. When foraging within fjords, birds are exposed to lower degrees of salinity due to freshwater inputs, decreased light penetration at deeper depths, unique bathymetry, and a lack of the usual open sea oceanographic variables (*Gibbs, 2001*; *Miller, Wing & Hurd, 2006*; *Wing & Jack, 2014*).

Past research has suggested that Milford Sound may be able to buffer fjord-based tawaki from the usually detrimental effects of an El Niño year and provide them with a productive alternative foraging environment (*Mattern & Ellenberg, 2016*). Utilising fjord-based habitats for foraging could be vital for the future survival of the species if marine perturbations and extreme weather events continue to increase in frequency. This study presents the first comparison of at-sea movements and dive behaviour between tawaki from two colonies from different locations within Piopiotahi/Milford Sound. The aim was to determine differences in the foraging strategies of an inner-fjord colony (Harrison Cove) and an outer-fjord colony (Moraine) over the years of 2019 and 2020. Milford Sound is a fjord of 16 km from entrance to head and the two colonies are separated by approximately

eight km. During guard-stage, female tawaki are the sole providers while males fast, remaining at the nest to guard their chick until it grows old enough to créche (*Warham, 1975*). Since only chick-rearing females were tracked, differences in dive behaviour are likely a result of individual/colony-based behavioural plasticity in response to the different environments in which birds foraged. Furthermore, foraging preferences of tawaki may change depending on the environmental and biological conditions they are faced with across years. As such, penguins breeding in outer fjord colonies should have the choice of either foraging inside or outside the fjord depending on foraging conditions, whereas inner fjord penguins must respond to the fjord environment presented to them each year and are more likely to forage exclusively inside the fjord, thereby avoiding longer travels to oceanic foraging grounds.

## METHODS

Portions of this text were previously published as a print of a research thesis (https://ourarchive.otago.ac.nz/esploro/outputs/graduate/Inter-annual-and-inter-colony-variation-in-the/9926479516601891).

### Study area

This study was conducted at Piopiotahi/Milford Sound, Fiordland (44.6414°S, 167.8974°E), a fjord on the southwest of New Zealand's South Island. This fjord hosts multiple colonies of tawaki with an estimated total population of 130–150 breeding pairs (*Mattern & Long, 2017*). Individuals were tracked at two sites: Harrison Cove, an inner-fjord colony and Moraine, an outer-fjord colony (Fig. 1). The Harrison Cove colony lies near the mouth of the Harrison River while the Moraine colony is situated near the fjord entrance opposite Dale Point. Harrison Cove hosts a breeding colony of roughly 16–20 breeding pairs, whereas the Moraine Colony is larger, hosting around 50 breeding pairs, ~25 of which were found in the section of the colony sampled.

### Deployment and retrieval of data loggers

Research took place over two consecutive breeding seasons, from late-September to mid-October in 2019 and mid-September to mid-October in 2020. GPS dive loggers were deployed on adult females during the chick guard stage of breeding. Tawaki are very secretive birds that breed in often hard to reach burrows (*Mattern & Wilson, 2018*). This fact combined with the relatively low number of nests per colony meant that sample sizes were inevitably lower than in comparable studies with species that breed in the open. On their return from foraging, and after allowing them time to feed their chicks and rest, adult females were captured at the nest by hand, or with the aid of a leg crook. Weight and beak measurements were taken to confirm sex (*White et al., 2021*). To quantify their foraging movements, and diving behaviour, birds were equipped with GPS dive loggers (Axy-Trek Marine, Technosmart) recording a location every 1–3 min, depth ($\pm$10 mbar of pressure), temperature ($\pm$ 0.2 °C) every 1 s, and tri-axial body acceleration at 25 Hz (40 × 20 × 8 mm, 14 g). The devices were attached to the midline dorsal feathers on the lower back with waterproof tape (TESA 4651; Beiersdorf AG) following *Wilson et al.*
*(1997)*. Attached devices represent well under 1% of the penguins' body mass in air and are streamlined to reduce drag, therefore they are likely to have negligible impact on the individual's foraging behaviour (*Agnew et al., 2013*). After 4–5 days the individuals were recaptured, and the data loggers removed without affecting plumage of the birds. This field study was approved by the University of Otago Animal Ethics Committee (AUP69-2017) and the New Zealand Department of Conservation Wildlife Authority (78612-FAU).

## Spatial data projection

ArcGIS release 10.3 was used to display foraging tracks for both Harrison and Moraine colonies over 2019 and 2020 (ESRI, Redlands, CA, USA). The bathymetric dataset "New Zealand Regional Bathymetry (2016)" was provided by NIWA and licensed under a Creative Commons Attribution-NonCommercial 4.0 International License (https://creativecommons.org/licenses/by-nc/4.0/). The coastlines base map was sourced from Toitū Te Whenua Land Information New Zealand (LINZ) and licensed by LINZ for re-use under the Creative Commons Attribution 4.0 International licence (https://data.linz.govt.nz/license/attribution-4-0-international/). The acquisition of a GPS fix takes 25–30 s. Consequently, some foraging tracks may appear to cross over land in cases where the device was unable to retrieve a location fix while tawaki were diving underwater and spending only short intervals at the surface.

## Data extraction and processing

With a custom-written script in MATLAB 9.9 (Mathworks Inc., Natick, MA, USA), GPS location data was used to calculate foraging parameters, including maximum distance from the colony per trip, cumulative distance travelled per trip, horizontal traveling speed, and trip duration (time between the first and last dive). To target foraging behaviour, only location points associated with dive events were used.

Dive data was also processed in MATLAB 9.9 to identify individual dive events and calculate parameters such as dive duration, descent duration and velocity, bottom time (*i.e.,* time spent at depth between descent and ascent), number of wiggles per dive (*i.e.,* the number of vertical undulations during the bottom phase, used as measure of foraging activity), maximum depth, ascent duration and velocity, and surface time. Following the methodology of *Ydenberg & Clark (1989)*, foraging efficiency was calculated as bottom time/(dive duration + surface time). Dives shorter than 5 s or shallower than 0.5 m were excluded to filter out non-foraging events or random pressure fluctuations. The 0.5 m threshold was established to allow us to capture shallow foraging dives made by fjord-foraging penguins, who tend to dive more shallowly (*Mattern & Ellenberg, 2019*; *Mattern & Ellenberg, 2016*). Other studies conducted on penguins both smaller and larger than tawaki, have accepted dives ≥0.5 m for analysis (*Ropert-Coudert, Chiaradia & Kato, 2006*; *Elley et al., 2022*). Due to analytical feasibility and the nature of the research questions, accelerometer data was omitted from analysis in the current study.

From the period of 10 September–15 October in both 2019 and 2020, hourly environmental data were sourced from a local weather station or monitoring buoy in Milford Sound and averaged to allow comparison across the years. The time period chosen

represents a core period of the tawaki breeding season and encompasses the days in which tawaki were actively tracked over both years. Daily measurements of total rainfall and average wind speed were collated from the Milford Sound Electronic Weather Station on NIWA's Cliflo database (https://cliflo.niwa.co.nz). Daily measurements of the temperature and salinity from sensors at an oceanographic mooring in the Milford Marina, were provided by Meridian Energy. The sensor data at depth 0.5 m were chosen for use in analysis of temperature and salinity as these give the most precise level of how thin/saline the low salinity layer is, and because 0.5 m was the shallowest accepted dive from tawaki in dive data analyses. For ocean-foraging birds, satellite data (see Table S2) was used to obtain seafloor bathymetry, daily sea surface temperature (SST), salinity, and chlorophyll-a, which were then aligned with tawaki GPS points and extracted. This resulted in a dataset of oceanic conditions encountered by tawaki across 2019 and 2020.

## Statistical analysis

The open source statistical software, R was used to perform data analysis and produce figures (*R Core Team, 2024*). Generalized linear mixed models (GLMMs) and linear mixed models (LMMs) were applied to compare foraging and dive parameters between colony sites (Moraine/Harrison Cove) and years (2019/2020). LMMs were fitted using the 'lmer' function in the 'lme4' package (*Bates et al., 2015*). GLMMs were specified using the 'glmer' function in 'lme4' or by using the 'glmmTMB' package, as it is equipped to handle more complex nested effect structures (*Brooks et al., 2017*). For foraging parameters, bird identity was included as a random factor to account for individual differences between birds, some of whom completed multiple foraging trips. For dive data, trip identity was nested within bird identity as some birds had dive data collected from multiple foraging trips. The year, colony and their interaction were fixed effects.

Prior to modelling, trip duration was log 10 transformed and cumulative distance travelled was square root-transformed to improve normality. GLMMs were used when LMMs violated the assumptions of normality (assessed with Shapiro–Wilk tests and residual plots). GLMMs were specified with the link function 'log' and the appropriate error structures (gamma for maximum distance, negative binomial for count data such as dives per hour and wiggles, Gaussian for horizontal speed and most dive parameters; refer to Table S3). Dispersion tests and residual checks were performed for each model using the 'DHARMa' package (*Hartig, 2024*). Type III Wald chi-square tests were used to determine the significance of fixed effects and interactions. Sum contrasts were set for categorical variables (colony and year) to ensure equal treatment of all factor levels in the analysis.

To compare environmental factors, LMMs and Type II Wald chi-square tests (for oceanic conditions) or ANOVA tests (for local fjord conditions) were used to assess significant differences between 2019 and 2020. For each LMM model, the residuals were plotted and visually checked to ensure there were no signs of abnormal patterns or heteroscedasticity.

## RESULTS

### Comparison of foraging behaviour between colonies and years

In 2019, foraging data were collected from six Moraine birds and four Harrison Cove birds, with a total of seven (out of 10) and five (out of seven) complete trips, respectively. In 2020, foraging data were collected from five Moraine birds and eight Harrison Cove birds, with a total of 13 (out of 16) and 18 (out of 20) complete trips, respectively. A complete trip refers to a foraging bout for which a full GPS track was recorded from when the bird left its colony to forage, until its return.

In 2019, 405 GPS locations from tracks associated with dive data were obtained for Harrison Cove birds and 1,547 from Moraine birds. In 2020, 1,033 and 2,840 foraging track locations were gathered from Harrison Cove and Moraine birds, respectively. Tracks revealed that in 2019, two of the four Harrison Cove birds exited the fjord to forage in the ocean whereas in 2020 only one of the eight birds tracked exited the fjord (Fig. 2). For Moraine, birds appeared to travel further from their colony, into the ocean during both years, although most foraging activity was concentrated closer inshore in 2020 (Fig. 2).

Multiple foraging trip parameters were significantly different across years and between colonies. In 2019, birds from both Moraine and Harrison Cove went on significantly longer foraging trips than in 2020. Furthermore, the trips made by Moraine birds were significantly longer than trips made by birds from Harrison Cove (Table 1). No birds from Harrison Cove in 2019 made multi-day foraging trips. Two of the four 2019 Harrison Cove birds foraged in the fjord, while the other two foraged outside, resulting in only 40% of complete trips from Harrison Cove occurring within the fjord, as compared to almost 90% of trips being fjord-based in 2020. Birds from both colonies travelled significantly greater maximum distances from their respective colony sites per trip in 2019 than in 2020, and Moraine birds travelled further from their colony than Harrison Cove birds (Table 1, Fig. 2). The interaction between colony and year was found to have a significant effect on the cumulative distance travelled, with birds from both colonies covering greater distances in 2019, and birds from Moraine generally travelling further than Harrison Cove tawaki (Table 1). There was a slight but significant effect of the year on the horizontal travelling speed of tawaki with birds moving at faster speeds in 2019 than in 2020 (Table 1). Overall, the most notable differences in foraging behaviour occurred between Moraine and Harrison Cove birds in 2020, and between Harrison Cove birds across the years.

### Comparison of dive behaviour between colonies and years

A total of 10,109 diving events were recorded for Moraine tawaki in 2019, and 9,154 in 2020. For Harrison Cove tawaki, a total of 5,832 dive events were recorded in 2019, and 7,573 in 2020. The interaction between the year and colony was significant for maximum dive depth and descent velocity and trending close to significance for dive duration (Table 2). On average, birds from the Moraine colony had longer dive durations in 2020 than in 2019, whereas birds from Harrison Cove dived for shorter periods of time in 2020 compared to 2019 (Table 2). Similarly, birds from Moraine dived to significantly greater depths and with faster descent rates in 2020 than in 2019, whereas the birds from Harrison Cove dived deeper and descended faster in 2019 than in 2020 (Table 2). Harrison Cove birds had

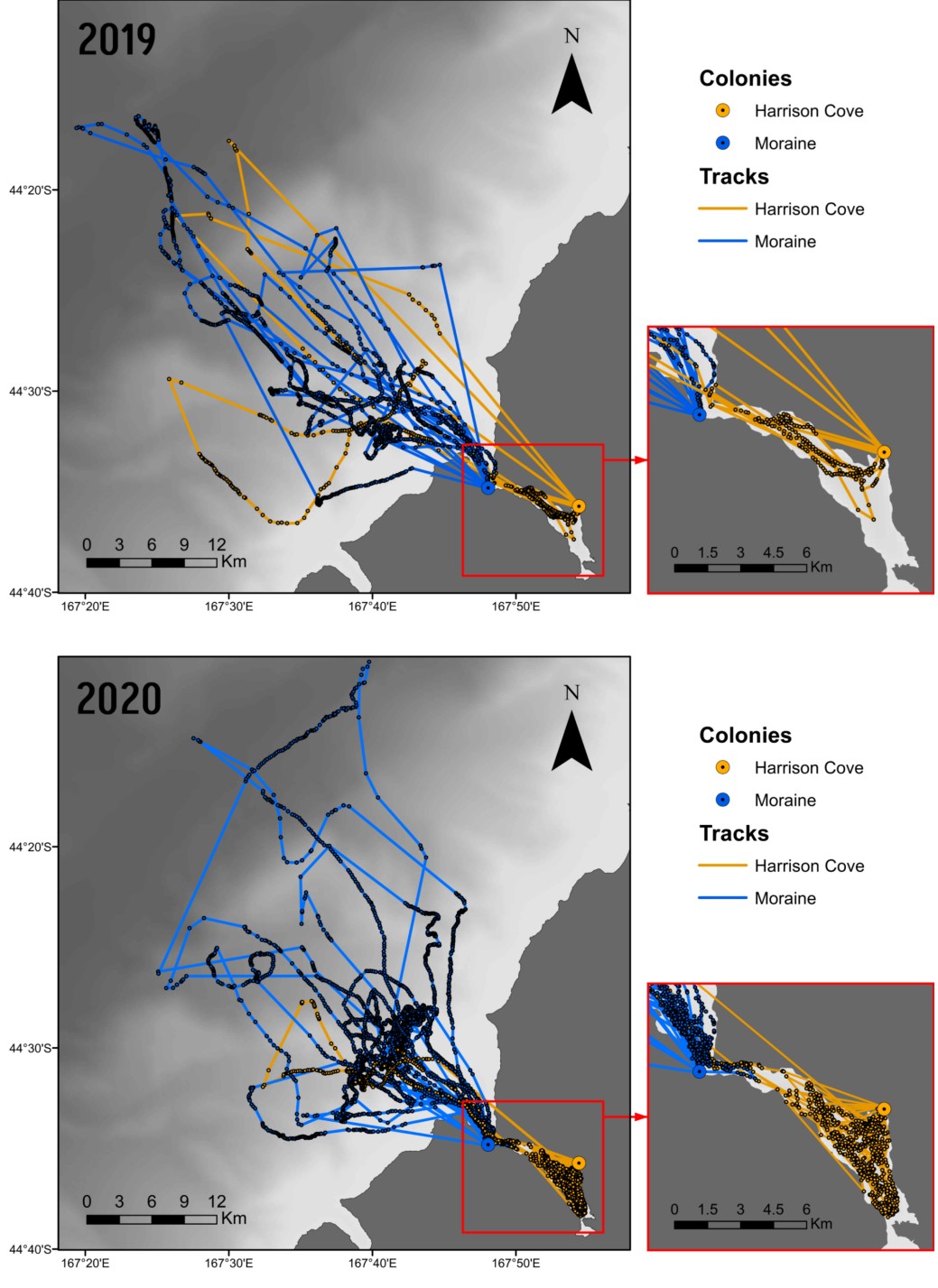

**Figure 2** **GPS points and foraging tracks of guard stage tawaki from the Harrison Cove and Moraine colonies in Milford Sound.** Birds tracked from the 2019 ($n = 10$) and 2020 ($n = 13$) breeding seasons. Fiord area is outlined (right). Shading represents the gradient of shallow (light) to deeper (dark) bathymetry. Base map data courtesy of LINZ, bathymetric data courtesy of NIWA.

**Table 1  Summary of the foraging attributes of tawaki tracked by year and colony.** Foraging parameters are displayed with their mean ± standard deviation, and corresponding results of the ANOVA test performed for each model. Degrees of freedom were 1,19. $X^2$ chi-square statistic and associated $p$ value also provided. $P$ values are in bold when under the set significance level of <0.05.

| | 2019 | | 2020 | | Predictors | $X^2$ value | $p$ value |
|---|---|---|---|---|---|---|---|
| | Moraine | Harrison Cove | Moraine | Harrison Cove | | | |
| Number of birds | 6 | 4[a] | 5 | 8 | – | – | – |
| Number of full trips | 7 | 5 | 13 | 18 | – | – | – |
| Number of trips >2 h | 4 | 2 | 4 | 0 | – | – | – |
| Percentage of trips in fjord only (%) | 0 | 40 | 6 | 89 | – | – | – |
| **Foraging parameters** | | | | | | | |
| Trip duration (h) | 31 ± 14 | 22 ± 11 | 25 ± 13 | 11 ± 4 | Colony | 9.33 | **0.002** |
| | | | | | Year | 5.84 | **0.016** |
| | | | | | Colony*Year | 1.99 | 0.158 |
| Max. distance from colony per trip (km) | 29.6 ± 9.2 | 26.3 ± 21.1 | 23.5 ± 13.2 | 7.7 ± 9.0 | Colony | 9.68 | **0.002** |
| | | | | | Year | 9.32 | **0.002** |
| | | | | | Colony*Year | 3.36 | 0.067 |
| Distance travelled per trip (km) | 74.4 ± 28.1 | 66.1 ± 40.8 | 66.5 ± 31.8 | 19.5 ± 10.7 | Colony | 8.34 | **0.004** |
| | | | | | Year | 7.69 | **0.006** |
| | | | | | Colony*Year | 4.27 | **0.039** |
| Travel speed (km h$^{-1}$) | 0.94 ± 0.10 | 1.00 ± 0.21 | 0.82 ± 0.10 | 0.87 ± 0.20 | Colony | 0.67 | 0.411 |
| | | | | | Year | 4.14 | **0.042** |
| | | | | | Colony*Year | 0.00 | 0.952 |

**Notes.**
[a] Data from five 2019 Harrison Cove birds were available for the parameter of trip duration (h).

faster descent rates and dived significantly deeper than Moraine birds in 2019, but this reversed in 2020 (Table 2). The number of wiggles per dive and the amount of time spent in the bottom phase of the dive was significantly higher in 2020 than in 2019, particularly for Moraine birds, indicating greater foraging activity in 2020 (Table 2). There were no significant differences between the years or colony sites in the number of dives per hour or the ascent rate (Table S3).

## Dive depth

In 2020, 85% of dives made by Harrison Cove birds were in the upper 0–20 m of the water column, whereas 51% of dives made by Moraine birds were within this depth range (Fig. 3). In comparison, Moraine birds in 2019 made 70% of their dives to maximum depths of 0–20 m, whereas in 2019 Harrison Cove birds made shallow dives less often, with only 58% of their maximum dive depths falling within 0–20 m (Fig. 3). Moraine birds dived deeper more often in 2020 compared to 2019, with 35% of their 2020 dives reaching depths of over 50 m, whereas just 10% of their 2019 dives were greater than 50 m (Fig. 3). For Harrison Cove, the trend was the opposite where about 3% of their dives were deeper than 50 m in 2020, but 22% of dives reached depths greater than 50 m in 2019 (Fig. 3).

In 2019, the distribution of dive depths throughout the day was skewed to the right with dives increasing in depth from 9 am onwards for Harrison Cove birds, peaking at maximum depth by 4 pm and decreasing in depth into the evening (Fig. 3). Dive depths

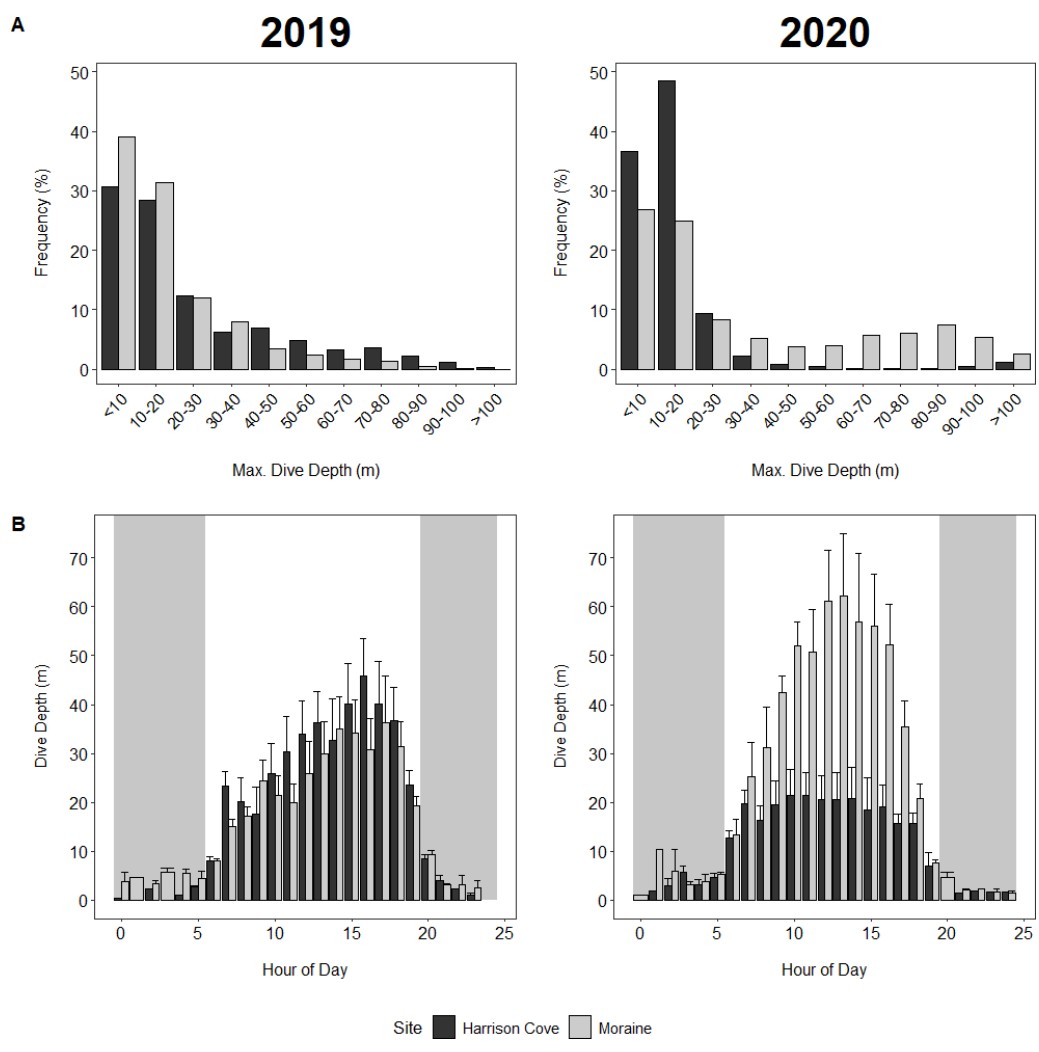

**Figure 3** **Dive behaviour of guard stage tawaki from two colonies in Milford Sound.** (A) Maximum dive depths for tawaki from Moraine and Harrison Cove during the guard stage of the 2019 (left) and 2020 (right) breeding seasons. (B) Average maximum dive depths (±standard error) across the hours of the day for tawaki from the Moraine and Harrison Cove colonies in Milford Sound during the guard stage of the breeding season in years 2019 (left) and 2020 (right). Grey areas represent night-time. Harrison Cove: $n = 5$ (2019), $n = 8$ (2020); Moraine: $n = 6$ (2019), $n = 5$ (2020).

throughout the day for Moraine birds in 2019 followed a similar pattern of peaking around late afternoon, although it was more erratic and less pronounced (Fig. 3). In 2020, however, Moraine birds had a highly symmetrical distribution where dives steadily increased in depth from 5 am and peaked between 12 pm–3 pm before becoming increasingly shallower by each hour into the evening (Fig. 3). In 2020, the hourly average depth distribution for Harrison Cove birds was almost uniform throughout the entire day from 6 am to 6 pm while foraging within the fjord. This was clearly different to how they foraged in 2019, as well as to Moraine birds (Fig. 3).

**Table 2 Summary of the dive parameters of tawaki tracked by year and colony.** Comparison of mean ± standard deviations of dive parameters for tawaki from the Moraine and Harrison Cove colonies in Milford Sound during the guard stage of the breeding season in years 2019 and 2020. The corresponding results of ANOVA tests performed for each model are also provided with an $X^2$ chi-square statistic and associated p values. P values are in bold when under the set significance level of $<0.05$; df = 1.

| | 2019 | | 2020 | | Predictors | $X^2$ value | p value |
|---|---|---|---|---|---|---|---|
| | Moraine | Harrison Cove | Moraine | Harrison Cove | | | |
| Number of birds | 6 | 5 | 5 | 8 | – | – | – |
| **Dive parameters** | | | | | | | |
| Dive duration (s) | 64.4 ± 17.0 | 71.1 ± 20.5 | 89.0 ± 19.3 | 66.7 ± 12.4 | Colony | 1.09 | 0.297 |
| | | | | | Year | 2.15 | 0.143 |
| | | | | | Colony*Year | 3.81 | 0.051 |
| Max. dive depth (m) | 19.0 ± 5.7 | 23.0 ± 7.8 | 30.9 ± 8.7 | 16.1 ± 5.2 | Colony | 3.57 | 0.059 |
| | | | | | Year | 0.28 | 0.594 |
| | | | | | Colony*Year | 10.91 | **<0.001** |
| Descent velocity (ms$^{-1}$) | 0.86 ± 0.11 | 0.90 ± 0.16 | 1.05 ± 0.08 | 0.85 ± 0.07 | Colony | 3.77 | 0.052 |
| | | | | | Year | 4.07 | **0.044** |
| | | | | | Colony*Year | 12.09 | **<0.001** |
| Foraging efficiency | 0.37 ± 0.03 | 0.38 ± 0.02 | 0.40 ± 0.03 | 0.43 ± 0.03 | Colony | 3.36 | 0.067 |
| | | | | | Year | 5.31 | **0.021** |
| | | | | | Colony*Year | 1.33 | 0.334 |
| Number of wiggles per dive | 7.6 ± 2.2 | 7.2 ± 2.8 | 9.2 ± 1.8 | 8.9 ± 1.4 | Colony | 0.14 | 0.706 |
| | | | | | Year | 7.66 | **0.006** |
| | | | | | Colony*Year | 0.08 | 0.777 |
| Bottom time (s) | 30.5 ± 8.7 | 32.3 ± 9.5 | 40.5 ± 8.9 | 35.2 ± 4.3 | Colony | 0.21 | 0.645 |
| | | | | | Year | 4.64 | **0.031** |
| | | | | | Colony*Year | 0.94 | 0.334 |

**Foraging efficiency**

The year significantly influenced foraging efficiency which was higher in 2020 than in 2019 (Table 2). For the Moraine colony, foraging efficiencies varied noticeably throughout the day, although the nature of this variation was different between years (Fig. 4). In 2020, the foraging efficiencies of Moraine birds decreased from 6 am to 2 pm, indicating increasingly long surface times between dives, before efficiencies generally increased again between 2 pm–5 pm (Fig. 4). The foraging efficiencies of Moraine birds in 2019 decreased from 6 am to 8 am and remained relatively constant throughout the morning-afternoon before rising again in the evening (Fig. 4). The foraging efficiencies of Moraine birds in 2019 were noticeably lower in the morning (8 am–11 am) compared to 2020. For Harrison Cove birds in 2019, foraging efficiencies decreased only from the early morning until 10 am and remained relatively constant through midday to evening while in 2020 Harrison Cove birds maintained a higher stable level of foraging efficiency throughout the daylight hours (6 am–7 pm) (Fig. 4).

Moraine birds tended to make shallower dives and have higher foraging efficiencies during the early morning and late evening of their foraging trips as they commuted to foraging grounds outside the fjord before making deeper dives focusing on feeding

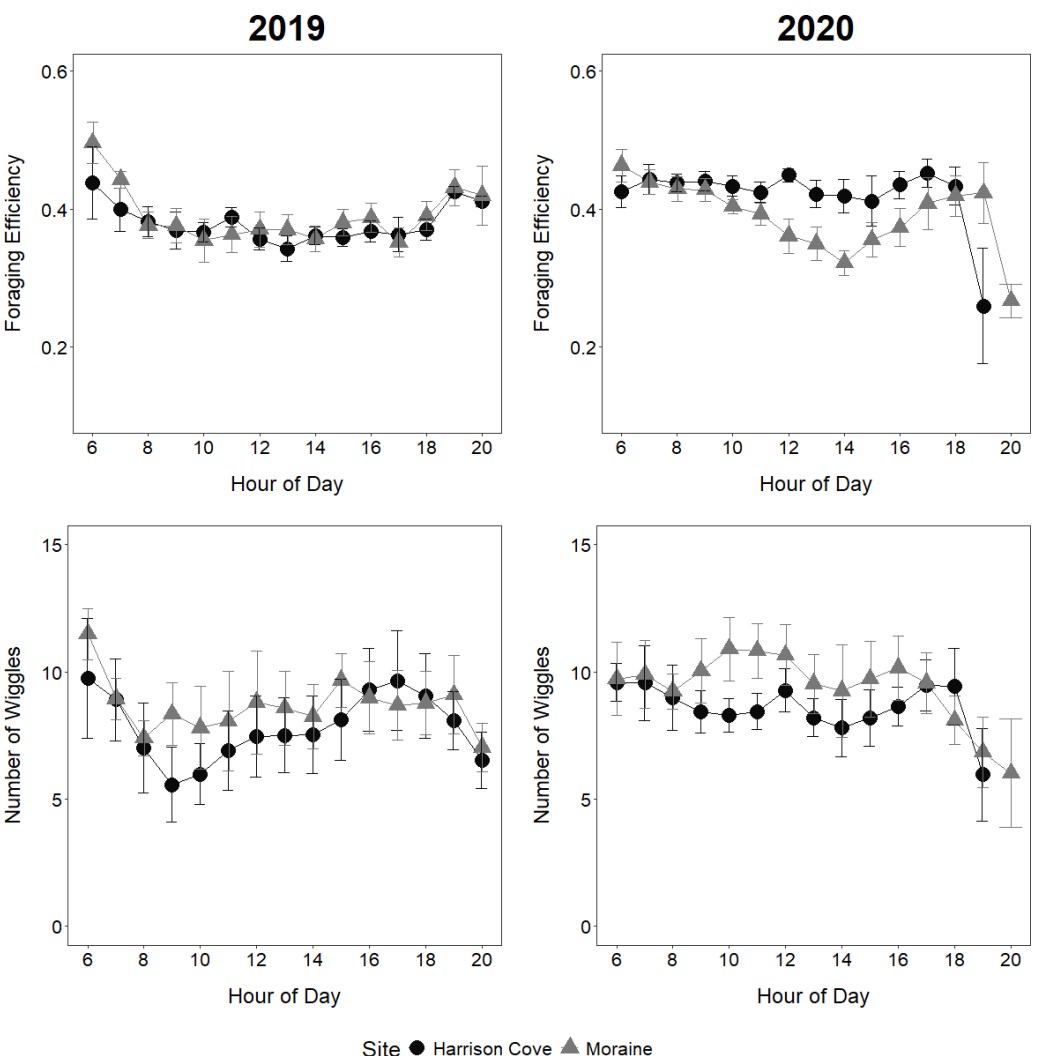

**Figure 4  Foraging efficiencies and number of wiggles per dive of guard stage tawaki from two colonies in Milford Sound.** Comparison of mean foraging efficiencies (±standard error) and mean number of wiggles per dive (±standard error) at each hour of the day for tawaki from two colonies in Milford Sound (Moraine and Harrison Cove) during the guard stage of the 2019 and 2020 breeding seasons. Harrison Cove: $n = 5$ (2019), $n = 8$ (2020); Moraine: $n = 6$ (2019), $n = 5$ (2020).

(Figs. 3 and 4). In contrast, dive depths and foraging efficiencies remained highly consistent for the 2020 Harrison Cove birds exclusive of time of day as they did not travel out to the open ocean (Figs. 3 and 4).

## Number of wiggles

The year significantly influenced the number of wiggles tawaki made per dive which was higher in 2020 than in 2019 (Table 2). In 2020, Moraine birds appeared to have their most productive foraging period between 9–11 am where their dives were most efficient and involved the greatest number of wiggles, indicating greater foraging activity (Fig. 4). For Harrison Cove birds in 2020, the number of wiggles per dive maintained

a relatively consistent trend throughout the day, with a peak at 12 pm, similar to their foraging efficiencies (Fig. 4). In contrast, dives made by Harrison Cove birds in 2019 had considerably lower numbers of wiggles on average, with a trough occurring at 8 am before increasing gradually throughout the day until 5 pm (Fig. 4). Dives made by Moraine birds in 2019 had more consistent numbers of wiggles across the day although these numbers were lower than their dives in 2020 (Fig. 4). The average number of wiggles per dive for tawaki from Moraine tended to be higher than that of tawaki from Harrison Cove over the core foraging hours of 9 am–3 pm in 2019 and 9 am–4 pm in 2020 (Fig. 4), however this trend was not significant (Fig. 4, Table 2).

### Conditions in the fjord environment in 2019 and 2020

Between the period of 10 September to 15 October, the mean temperature and mean salinity of the fjord waters in Milford Sound at 0.5 m depth were significantly lower in 2020 compared to 2019 (ANOVA, $F = 35.58$, $p < 0.001$; ANOVA, $F = 36.42$, $p < 0.001$; Fig. 5). The mean daily rainfall during this period appeared higher in 2020 compared to 2019, although this difference was not statistically significant (ANOVA, $F = 3.22$, $p = 0.077$; Fig. 5). Mean wind speed also appeared to be marginally higher in 2020 than in 2019, however, this difference was not statistically significant (ANOVA, $F = 2.99$, $p = 0.089$; Fig. 5).

### Conditions in the ocean environment in 2019 and 2020

The oceanic environments and conditions encountered by tawaki foraging outside Milford Sound varied considerably between 2019 and 2020 (Table S4; Fig. 6). The mean SST encountered by tawaki while foraging in the ocean outside Milford Sound was significantly higher in 2020 compared to 2019 (LMM, $X^2 = 3,210$, $p < 0.001$; Fig. 6). The salinity of the ocean over which birds foraged was significantly, although only slightly, lower in 2020 than in 2019 (LMM, $X^2 = 30.82$, $p < 0.001$; Fig. 6). Mean chlorophyll-a concentration was significantly lower in 2020 than 2019 (LMM, $X^2 = 8,395$, $p < 0.001$; Fig. 6). Tawaki also foraged in waters that were of significantly shallower seafloor bathymetry in 2020 than in 2019 (LMM, $X^2 = 27.96$, $p < 0.001$; Table S4; Fig. 6).

## DISCUSSION

Tawaki exhibited inter-annual and inter-colony differences in diving and foraging patterns. Contrary to our initial predictions, penguins from the outer fjord always foraged in the ocean outside, whereas birds from the inner fjord colony showed greater plasticity in their foraging behaviour by showing similar oceanic foraging in 2019 but foraged almost exclusively inside the fjord in 2020. This plasticity in foraging strategy by inner fjord birds may be an advantage in an era of rapid changes to the environment.

### Differences in foraging behaviour between colonies and years

Tawaki showed clear polarity in their spatial use and foraging trip parameters between colonies, with the most striking differences occurring in the 2020 breeding season when almost all Harrison Cove tawaki foraged within the fjord while Moraine birds foraged

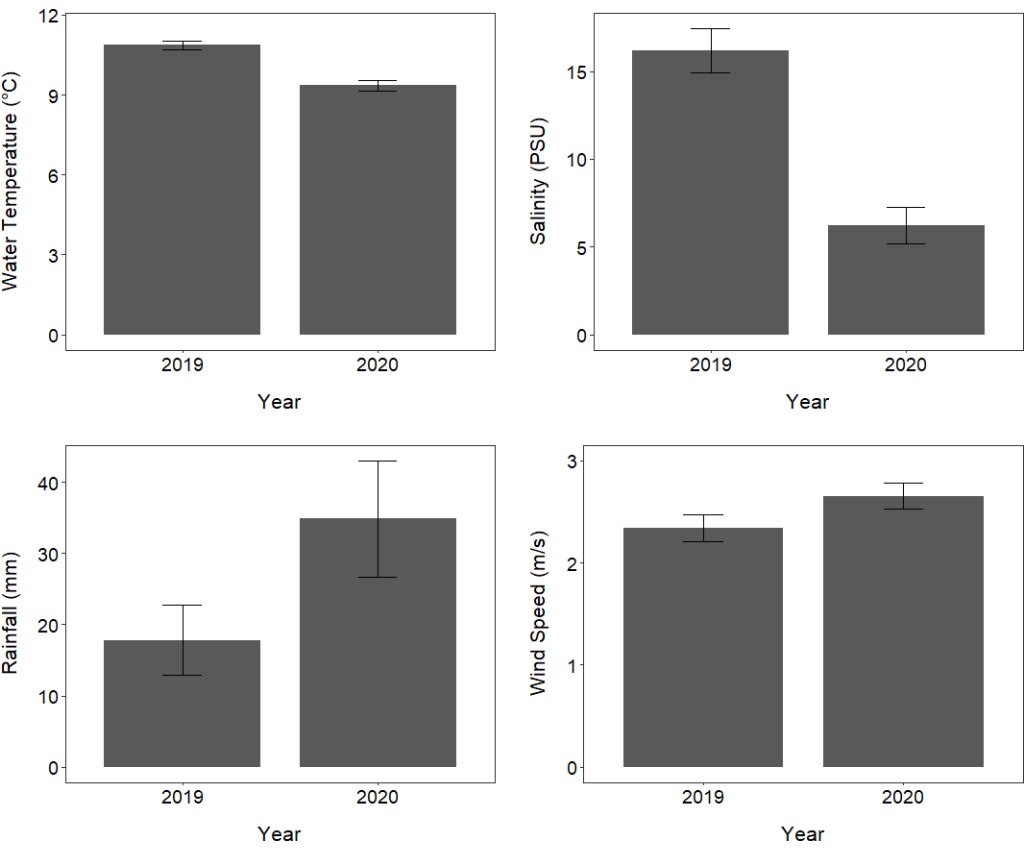

**Figure 5** **Environmental conditions in Milford Sound over two consecutive years.** The mean (±standard error) of daily rainfall and wind speed as well as water temperature and salinity at 0.5 m depth in Milford Sound throughout the core tawaki breeding season (10 September –15 October) in the years 2019 and 2020.

in the open ocean. This is consistent with the common phenomenon of inter-colony segregation, observed in penguins as well as multiple other seabird species, to mitigate intra-specific competition (*Hoskins et al., 2008*; *Bolton et al., 2019*; *Gulka et al., 2020*). Although there appears to be a high prevalence of inter-colony segregation in seabirds, complete segregation within obvious boundary lines is rare and usually there is some degree of partial overlap (*Bolton et al., 2019*; *Ito et al., 2021*). Such overlap occurred for one to two Harrison Cove birds who made foraging trips out into the open ocean, where Moraine birds also foraged. Notably, there was only one instance of a Moraine bird foraging within the outer fjord area and zero cases of Moraine birds foraging in the inner fjord, within 10 km of the fjord head, suggesting a potential foraging boundary in the inner fjord.

No Moraine birds foraged extensively within the fjord even though it appears to be a productive foraging habitat, as indicated in 2020 by Harrison Cove birds that had significantly shorter trip durations, and foraging distances. Initially, we thought that tawaki from a colony at the fjord entrance would show greater flexibility in foraging as they have immediate access to both oceanic and fjord environments. Instead, it was the inner fjord

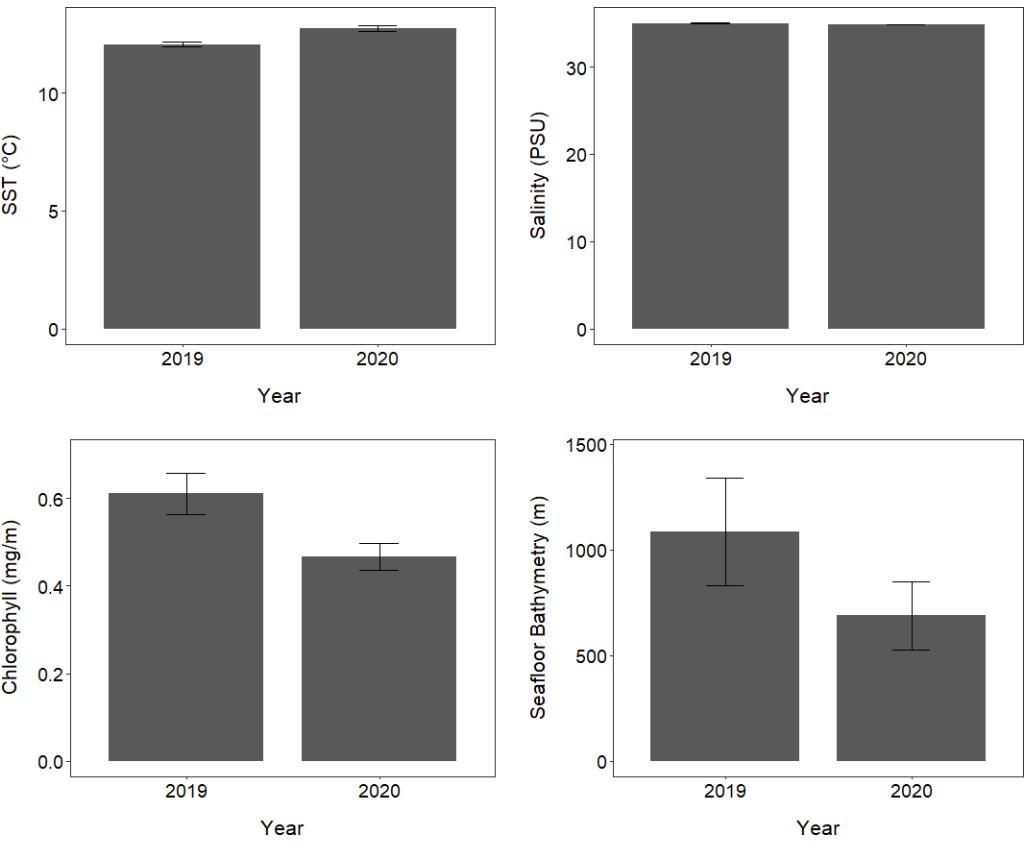

**Figure 6  Oceanic conditions outside Milford Sound over two consecutive years.** The mean (±standard error) of environmental conditions experienced by tawaki foraging in the ocean outside Milford Sound fiord, in the years of 2019 ($n = 6$) and 2020 ($n = 5$).

birds that adjusted their behaviour to either forage outside the fjord (2019), presumably because foraging conditions inside weren't favourable, or to remain inside when they were (2020). This, in turn, seems to suggest that foraging strategies in chick rearing tawaki follow a simple rule—head towards the open ocean until you encounter adequate foraging conditions. Several marine animals exhibit flexibility in their foraging strategies despite initially moving offshore towards areas of open ocean (*Ladds et al., 2020*; *Jacoby Roy et al., 2025*). For example, harbour seals (*Phoca vitulina*) exploit locally abundant prey patches within fjords as they move toward the mouth (*Ramasco, Biuw & Nilssen, 2014*; *Wilson et al., 2014*). In years where foraging conditions are favourable inside the fjord, Harrison Cove penguins may be at an advantage over their Moraine counterparts.

In murres (*Uria aalge*) breeding at a large offshore and a small inshore colony, birds from the offshore colony travelled greater distances with more time spent commuting (*Gulka et al., 2020*). It was hypothesised that this cost was balanced by the benefit of reaching predictable and abundant aggregations of prey (*Gulka et al., 2020*). Considering the Moraine penguin colony comprises almost three times as many tawaki nests than Harrison Cove (*Mattern & Long, 2017*), it can certainly be argued that an exclusive oceanic

foraging strategy is not detrimental to the penguins' breeding success. Conversely, penguins from Harrison Cove may experience more extreme fluctuations in their breeding success considering that in 'poor' years like 2019, their commute to the ocean adds substantial energy expenditure to supplying their offspring with food. Yet, in 'good' years the opposite is likely the case. Furthermore, the presence of the Piopiotahi Marine Reserve may be a contributing factor in Milford Sound.

Fjord-foraging tawaki in 2019 and 2020 foraged on both sides of the fjord, however, there was often significant activity around Stirling Falls on the northern side—likely due to this feature being a rich source of freshwater input as well as being within the Piopiotahi Marine Reserve. Following the protection of inner fjord habitats in Fiordland, the abundance of omnivorous fish species increased, and the community stability of reef fishes also improved within reserves (*Wing & Jack, 2013*). Although the rest of Milford Sound is likely gaining spillover benefits from the marine reserve, there is still a considerable amount of fishing activity taking place in the unprotected zones, particularly at the fjord entrance (*Davey & Hartill, 2011*; *Fiordland Marine Guardians, 2021*). On the northern side of the fjord, Harrison Cove birds immediately enter a protected, unharvested environment upon leaving their colony, whereas on the southern side, Moraine birds enter unprotected waters subject to greater oceanic influence. This difference in environment may drive Moraine birds towards the ocean, consequently forgoing feeding opportunities within the fjord despite its potential productivity.

At 16 km, Milford Sound is the shortest fjord and given its length, oceanic foraging remains an option for inner fjord tawaki. However, for tawaki from colonies in longer fjords, such as those 40 km from the entrance in Doubtful Sound (*Mattern, 2013*; *Hornblow, 2022*), exclusive oceanic foraging may be impractical, as the probability of chick mortality tends to increase with trip distance, particularly at the beginning of guard stage, when chicks are very young (*Boersma & Rebstock, 2009*). This indicates that fjord-bound birds are able to adjust their foraging behaviour in different ways to what we observed in Harrison Cove. The variability of exclusive fjord foraging strategies is currently being analysed in Doubtful and Dusky Sound.

### Differences in dive behaviour between colonies and years

In 2019, birds from Harrison Cove were diving significantly deeper, for longer and with higher descent rates than birds from Moraine, while in 2020 these trends were reversed. Such shifts in dive behaviour by marine predators typically reflect changes in prey distribution and availability (*Kooyman et al., 1992*). Despite Moraine birds consistently foraging in the oceanic environment, their dive behaviour changed significantly between years, suggesting altered marine prey distributions, a pattern supported by prior shifts in tawaki diets during the ENSO event in 2015–2016 (*White, 2020*).

Diving to deeper depths was central to the foraging strategy of Moraine birds in 2020 and their foraging could be split between two approaches—diving shallowly in the upper 20 m of the water column or diving deeper past the 50 m mark. Increases in the frequency of dives deeper than 50 m for Moraine birds suggests a higher foraging effort at these depths. There have been cases where penguins that struggled to provide for their chicks dived
deeper and showed higher diving effort than conspecifics with shallower diving (*Chiaradia et al., 2007*). In 2020, fjord prey for Harrison Cove birds was likely plentiful in the upper 20 m of the water column, as this was where 85% of dives occurred. Although it seems that Moraine birds would expend more energy diving deeper and longer than Harrison Cove birds in 2020, their average foraging efficiencies were similar suggesting that the birds differed in diving preferences but did so within comparable physiological parameters. Furthermore, in 2020 we found no chick deaths due to starvation and both colonies had examples of nests raising two chicks. Despite tawaki being described as obligate brood reducers (*St Clair, 1992*), they have been known to raise more than one chick in years of favourable foraging conditions (*Mattern & Ellenberg, 2016*). This, in turn, suggests that La Niña conditions may be favourable for the species, which is in contrast to the other mainland penguin species, yellow-eyed penguin/hoiho and little penguin/kororā. Hoiho experience significant breeding failure during La Niña (*Mattern et al., 2017*) while kororā undergo detrimental changes to their onset of breeding (*Perriman et al., 2000*) and may be unable to provide sufficient food for their chicks during marine heatwaves (*Salinger et al., 2023*).

Unlike hoiho and kororā, which primarily forage in the shallow (<200 m) continental shelf waters close to mainland New Zealand (*Hickcox et al., 2022*; *Chiaradia et al., 2007*), tawaki inhabit a distinctly different marine environment. Their foraging grounds encompass quasi-pelagic conditions beyond the fjords, as well as the unique fjord ecosystems themselves. Notably, penguin species vulnerable to La Niña events are typically found along eastern coastlines, where onshore winds bring increased cloud cover and precipitation—factors that can dimmish light penetration and increase turbidity, negatively impacting visual foragers (*Cannell & Cullen, 2008*; *Darby et al., 2022*). During El Niño years, the west coast experiences comparable conditions such as higher precipitation and onshore winds (*Mullan & Thompson, 2006*), meaning tawaki are exposed to similar climatic challenges faced by eastern species, but linked to the opposite phase of the ENSO cycle.

## The variability of the oceanic foraging environment

As central-place foragers during the breeding season, tawaki are required to adapt their foraging behaviour in response to environmental variation. This was evident in Moraine birds who undertook longer, more distant trips in 2019 but dived more shallowly, whereas in 2020, trips were shorter and closer to the colony, but dives were consistently deeper. Based on these results, it is hard to determine whether one year was a "better" foraging year than the other. Studies on other penguins indicate that shallower foraging years represent more favourable foraging conditions, suggesting 2019 was the better year (*Ponganis et al., 2000*; *Berlincourt & Arnould, 2015*). However, taking the foraging trip parameters into account, there appeared to be more feeding opportunities closer to the colony in 2020, suggesting that this year was more favourable (*Burke & Montevecchi, 2009*; *Ramos et al., 2018*). Regardless, tawaki adapted to altered conditions in the oceanic environment by adjusting aspects of their foraging trips or dive behaviour to maximise their energy efficiency and successfully rear offspring in both years.

In 2020 the coastal ocean outside of Milford Sound was significantly warmer than in 2019 (Fig. 6), due to La Niña. While warmer SSTs have varied impacts on seabird foraging success (*e.g.*, *Pinaud & Weimerskirch, 2002*; *Cullen et al., 2009 versus Guinet et al., 1998*; *Ramos et al., 2018*), for tawaki, warmer water and decreased chlorophyll-a concentrations did not signify a less productive foraging environment. Prey abundance appeared to have shifted closer to the colony in 2020 and despite deeper dives, Moraine birds showed higher foraging efficiencies and more intense foraging activity (indicated by wiggles, a known proxy for prey encounters; *Simeone & Wilson, 2003*; *Bost et al., 2007*). As warmer SSTs result in increased stratification, it is possible that prey are concentrated at the thermocline (*Kitagawa et al., 2000*) and less dispersed in the water column (*Abookire, Piatt & Robards, 2000*), leading to more efficient foraging by marine predators (*Ropert-Coudert, Kato & Chiaradia, 2009*). The deeper dives of tawaki likely targeted thermally limited pelagic fish that moved to cooler, deeper waters below the thermocline (*Sabates et al., 2008*; *Santora et al., 2014*).

Furthermore, there have been instances in parts of New Zealand where in years of warmer ocean temperatures, there is enhanced survival of larval and juvenile fish species, as well as movement of deeper water species to more shallow coastal waters (*Basher, 1998*). Small or larval-stage schooling fish, winter-spawning squid, and krill are the main dietary components of tawaki (*Van Heezik, 1989*; *Van Heezik, 1990*; *Poupart et al., 2019*; *Hornblow, 2022*). Research from North America has demonstrated that definitive shifts in species composition of larval fish occur concurrently with shifts in zooplankton biomass, in relation to El Niño or La Niña (*Brodeur et al., 2008*; *Zamon & Welch, 2005*). In the warm phase of ENSO, it is more likely for larval species with a tropical-subtropical affinity to dominate (*Sánchez-Velasco et al., 2004*), although the effect of this on forage fish and higher trophic levels remains unclear (*Hill, Daly & Brodeur, 2015*). Diet analyses of tawaki and forage fish during La Niña, and in comparison to cooler oceanic conditions are necessary to understand prey species susceptibility to climatic change.

In 2020, tawaki spent more time diving in shallower neritic waters (56%) compared to 2019 (37%), where oceanic waters were preferred. This shift in bathymetric foraging preference, alongside potentially deeper prey aggregations in productive shallow coastal areas (*Miller, 2009*; *Chiaradia et al., 2007*), likely explains the deeper dives by Moraine birds in 2020. Local oceanography plays a significant role in tawaki foraging, as evidenced by minimal use of shelf-slope waters which are prominent around coastal tawaki colonies in other regions such as Open Bay Island (*Poupart et al., 2019*), but largely absent off Fiordland.

Moraine tawaki alternated their foraging strategies dependent on marine conditions across the years. In 2019 the ENSO was in a "normal" phase, while in 2020 the ENSO was in a La Niña phase. Birds from Moraine, while never foraging solely within the fjord in 2019 or 2020, do technically have the option to switch from ocean-foraging to fjord-foraging, for example, during an El Niño year. Although seabirds can be adept at altering their foraging behaviour to buffer environmental changes in prey distributions (*Litzow et al., 2002*; *Garthe, Montevecchi & Davoren, 2011*), there are eco-physiological limits to foraging plasticity (*Sommerfeld et al., 2015*). In addition to events like ENSO,
rising ocean temperatures and an increase in ocean acidity will likely lead to the seas of the Southern Ocean experiencing trophic shifts and changes to food supply for marine predators (*Ramírez et al., 2017*). If these changes to the oceanic environment cause prey availability to drop below a critical threshold, tawaki from Moraine might find open ocean foraging unprofitable. Such an outcome would echo the negative consequences on reproduction reported for tawaki at the Jackson Head colony, 80 km north of Milford Sound, during the severe 2015 El Niño (*Mattern & Ellenberg, 2016*), as well as in other seabird species (*Harding et al., 2007*; *Grémillet & Boulinier, 2009*). It remains to be seen whether birds from the Moraine colony will be at a disadvantage with the progressing changes in the world's oceans as a result of climate change (*Law et al., 2018*; *Sutton & Bowen, 2019*; *Behrens et al., 2022*). Conversely, having the option of foraging exclusively inside the fjord may be beneficial for penguins from Harrison Cove.

## The fjord as a unique foraging refuge

In 2020, Harrison Cove tawaki primarily used Milford Sound fjord, with an average maximum foraging distance of only eight km from their colony, a threefold reduction compared to Moraine birds. Fjord-based foraging, characterized by shorter trips and shallower dive behaviour, appears to be more energy-efficient than oceanic foraging. The deepest dive a Harrison Cove bird made in the fjord was 82 m, whereas outside the fjord a Harrison Cove bird dived to 113 m. Given that the deepest part of Milford Sound is 300 m, topography is not a limiting factor preventing fjord-foraging birds from diving deeper. However, New Zealand fjords do have a low underwater light environment due to a combination of topographic shading by steep fjord walls and the buoyant, tannin rich low salinity layer that attenuates light (*Grange et al., 1981*; *Grange & Singleton, 1988*; *Gibbs, 2001*). Since penguins are visual foragers (*Martin & Young, 1984*; *Cannell & Cullen, 2008*), it is likely that light conditions in the fjord dictate the depth to which tawaki can forage successfully. This aligns with the primarily daylight diving patterns of many penguin species (*Wilson et al., 1993*; *Walker & Boersma, 2003*), and those observed during this study. Deeper fjord dives (>50 m) were likely for predator evasion, not foraging, given the minimal light at these depths.

The conditions within Milford Sound were notably different between 2019 and 2020, with the fjord having significantly lower salinity and colder water temperatures as well as slightly higher wind speeds and rainfall in 2020. As tawaki from Harrison Cove foraged mainly in the fjord in 2020, but less so in 2019, it is possible that the 2020 conditions represented a more favourable fjord foraging environment. The higher rainfall and stronger winds in 2020 would have contributed to the resultant colder temperatures and lower salinity, representing a thicker low salinity layer (*Gibbs, 2001*) that was present more consistently in 2020. These conditions have the capacity to create a richer foraging environment as increased freshwater input can amplify estuarine circulation and increase nutrient input to the fjord, thereby enhancing productivity, which can have flow-on effects across the greater food web (*Rysgaard et al., 2003*). Higher rainfall in 2020 may have also led Harrison Cove birds to dive more shallowly, without sufficient light available to forage at greater depths.

It is not well known what makes New Zealand fjords such a productive foraging environment for marine predators, with most studies on the ecology of the upper trophic levels of fjord systems having been conducted in Scandinavian arctic fjords or Chilean fjords (*McMeans et al., 2013*; *Borras-Chavez et al., 2024*). In these systems it has been shown through biodiversity surveys that a plethora of novel species exist, where they may be at odds with their preferences elsewhere (*Försterra & Häussermann, 2003*; *Sinniger & Häussermann, 2009*). Fjords provide a substantial degree of habitat complexity to marine life, and many fish species are known to seek out areas that are structurally complex (*Chittaro, 2004*). In Fiordland, the outer reaches of fjords have been characterised as productive environments with higher light and wave energy allowing faster growth rates and fecundity of marine species, while the internal waters support recruitment of species from outside in a dynamic source–sink structure (*Wing, 2009*; *Wing & Jack, 2014*). Higher recruitment rates of prey species into inner fjord populations may boost foraging activity for predators like tawaki. New Zealand's fjord walls themselves are highly productive due to the accumulation of detritus and rockfall (*McLeod & Wing, 2007*), providing varied feeding opportunities, as seen with bottlenose dolphins in Doubtful Sound (*Bennington et al., 2021*).

The protection of fjords as refugia for tawaki is particularly important considering that adverse marine conditions and the frequency of El Niño events are expected to increase (*Sutton & Bowen, 2019*; *Cai et al., 2015*). The phenomenon of refugial habitats during El Niño has also been documented in the Canal de Ballenas in the Central Gulf of California where highly mobile marine animals left areas of reduced productivity and moved to the Canal during the 1983 El Niño as productivity there was high irrespective of the ENSO phase (*Tershy, Breese & Alvarez-Borrego, 1991*). *Aid, Montgomery & Mock (1985)* also reported that the Gulf of Panama may have been a high productivity refuge during the 1983 El Niño as SST there remained relatively cool and attracted high abundances of seabirds. While the current study's geographical scope is limited to one fjord, recent research in a much longer fjord revealed comparable patterns: outer-fjord tawaki foraged coastally with deeper dives, exhibiting no overlap of core foraging areas with mid-fjord tawaki, who mostly made shorter, shallower dives within the fjord (*Hornblow, 2022*). Fjords like Milford Sound clearly have the potential to act as a refuge to marine life during unfavourable foraging years, protecting tawaki from adverse conditions in the oceanic environment—which is good news in the face of current and predicted environmental change.

## CONCLUSIONS

This study revealed distinct foraging strategies between inner and outer fjord tawaki colonies. Contrary to predictions, Moraine (outer-fjord) birds consistently foraged in the open or coastal ocean, while Harrison Cove (inner-fjord) birds exhibited greater plasticity, showing a preference for fjord-foraging while making the occasional oceanic trip. We hypothesise that spatial foraging strategies in chick-rearing tawaki are determined by simply heading towards the open ocean until they encounter adequate foraging conditions. In years when foraging conditions are favourable inside the fjord, Harrison Cove penguins benefit

from shorter foraging trips with shallower diving. For Moraine penguins, although they did adjust their oceanic foraging and diving across the years, they may be at a disadvantage if they are unable to modify their behaviour to make use of the fjord in seasons of low productivity. Similar behavioural patterns have been observed in tawaki across most of the species' breeding range in previous (*Mattern & Ellenberg, 2016*; *Mattern & Ellenberg, 2017*; *Mattern & Ellenberg, 2018*) as well as subsequent years (*Hornblow, 2022*). Combined, these findings highlight the considerable foraging plasticity in tawaki. The behaviour of tawaki across the rest of New Zealand's fjord systems should be investigated in future to further understand the significance of fjords in the conservation of this species and their potential to act as foraging refugia.

## ACKNOWLEDGEMENTS

We are grateful to Southern Discoveries in Milford Sound for logistical support, in particular, Andrea Faris, Wolfgang Hainzl, Dan Crook, Joe Masters and Emma Thompson. We thank Eric Goodwin and Robyn Dunmore from Meridian Energy for data from the oceanographic monitoring buoy in Milford Sound. Special thanks also to the volunteers, Hannah Mattern, Deleece Augustyn, Lindsay Chan, Kolja Dorschel, Sylvain Zat and Braydon Moloney who assisted us during fieldwork.

### Funding

This work was supported by a Birds New Zealand Research Grant, and by patrons of the Tawaki Project on Patreon. GPS dive logger devices were purchased by the Global Penguin Society. The funders had no role in study design, data collection and analysis, decision to publish, or preparation of the manuscript.

### Grant Disclosures

The following grant information was disclosed by the authors:
Birds New Zealand Research Grant.
Tawaki Project on Patreon.
Global Penguin Society.

### Competing Interests

Yolanda van Heezik is an Academic Editor for PeerJ.

### Author Contributions

- Myrene Otis conceived and designed the experiments, performed the experiments, analyzed the data, prepared figures and/or tables, authored or reviewed drafts of the article, and approved the final draft.
- Thomas Mattern conceived and designed the experiments, performed the experiments, authored or reviewed drafts of the article, and approved the final draft.
- Ursula Ellenberg performed the experiments, authored or reviewed drafts of the article, and approved the final draft.

- Robin Long performed the experiments, authored or reviewed drafts of the article, and approved the final draft.
- Pablo Garcia Borboroglu conceived and designed the experiments, authored or reviewed drafts of the article, and approved the final draft.
- Philip J. Seddon conceived and designed the experiments, authored or reviewed drafts of the article, and approved the final draft.
- Yolanda van Heezik conceived and designed the experiments, authored or reviewed drafts of the article, and approved the final draft.

## Animal Ethics

The following information was supplied relating to ethical approvals (i.e., approving body and any reference numbers):

The study was approved by the University of Otago Animal Ethics Committee (AUP69-2017).

## Field Study Permissions

The following information was supplied relating to field study approvals (i.e., approving body and any reference numbers):

Field experiments were approved by the New Zealand Department of Conservation Wildlife Authority (78612-FAU).

## Data Availability

The dive data and analysis code are available at GitHub and Zenodo:

– https://github.com/Myrene-O/Milford-Sound-Tawaki/tree/main

– Otis, M. (2025). Inter-colony and inter-annual behavioural plasticity in the foraging strategies of a fjord-dwelling penguin (v1.0.0) [Data set]. Zenodo. https://doi.org/10.5281/zenodo.14849008.

The raw, unfiltered tracks and foraging movement data are available at Movebank: Otis et al. (in submission) Inter-colony and inter-annual behavioural plasticity in the foraging strategies of a fjord-dwelling penguin –good news in the face of environmental change? https://www.movebank.org/cms/webapp?gwt_fragment=page%3Dstudies%2Cpath%3Dstudy5596513373.

## Supplemental Information

Supplemental information for this article can be found online at http://dx.doi.org/10.7717/peerj.19650#supplemental-information.

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

# PeerJ

**Simeone A, Wilson RP. 2003.** In-depth studies of Magellanic penguin (*Spheniscus magellanicus*) foraging: can we estimate prey consumption by perturbations in the dive profile? *Marine Biology* **143**:825–831 DOI 10.1007/s00227-003-1114-8.

**Sinniger F, Häussermann V. 2009.** Zoanthids (Cnidaria: Anthozoa: Zoanthidae) from shallow water of the southern Chilean fjord region with the description of a new genus and two new species. *Organisms Diversity & Evolution* **9**:23–36 DOI 10.1016/j.ode.2008.10.003.

**Sommerfeld J, Kato A, Ropert-Coudert Y, Garthe S, Wilcox C, Hindell MA. 2015.** Flexible foraging behaviour in a marine predator, the Masked booby (*Sula dactylatra*), according to foraging locations and environmental conditions. *Journal of Experimental Marine Biology and Ecology* **463**:79–86 DOI 10.1016/j.jembe.2014.11.005.

**St Clair CCS. 1992.** Incubation behavior, brood patch formation and obligate brood reduction in Fiordland crested penguins. *Behavioral Ecology and Sociobiology* **31**:409–416 DOI 10.1007/bf00170608.

**Sutton PJH, Bowen M. 2019.** Ocean temperature change around New Zealand over the last 36 years. *New Zealand Journal of Marine and Freshwater Research* **53**:305–326 DOI 10.1080/00288330.2018.1562945.

**Tershy BR, Breese D, Alvarez-Borrego S. 1991.** Increase in cetacean and seabird numbers in the Canal de Ballenas during an El Niño-Southern Oscillation event. *Marine Ecology Progress Series* **69**:299–302 DOI 10.3354/meps069299.

**Trathan PN, Murphy EJ, Croxall JP, Everson I. 1998.** Use of at-sea distribution data to derive potential foraging ranges of macaroni penguins during the breeding season. *Marine Ecology Progress Series* **169**:263–275 DOI 10.3354/meps169263.

**Tremblay Y, Cherel Y. 2003.** Geographic variation in the foraging behaviour, diet and chick growth of rockhopper penguins. *Marine Ecology Progress Series* **251**:279–297 DOI 10.3354/meps251279.

**Van Heezik Y. 1989.** Diet of Fiordland crested penguins during the post-guard phase of chick growth. *Notorins* **36**:151–156 DOI 10.1080/03014223.1990.10422952.

**Van Heezik Y. 1990.** Diets of yellow-eyed, Fiordland crested, and little blue penguins breeding sympatrically on Codfish Island, New Zealand. *New Zealand Journal of Zoology* **17**:543–548 DOI 10.1080/03014223.1990.10422952.

**Walker BG, Boersma PD. 2003.** Diving behavior of Magellanic penguins (*Spheniscus magellanicus*) at Punta Tombo, Argentina. *Canadian Journal of Zoology* **81**:1471–1483 DOI 10.1139/z03-142.

**Warham J. 1975.** The crested penguins. In: Stonehouse B, ed. *The biology of penguins*. Baltimore: University Park Press, 189–269.

**White JW. 2020.** Foraging strategy plasticity in Fiordland Penguins (*Eudyptes pachyrhynchus*): a stable isotope approach. Dissertation, Marshall University.

**White JW, Mattern T, Ellenberg U, Garcia-Borboroglu P, Houston DM, Seddon PJ, Mays HL. 2021.** Field sexing techniques for Fiordland crested penguins (tawaki; *Eudyptes pachyrhynchus*). *Notornis* **68**:188–193 DOI 10.63172/405994cohhps.

**Williams TD, Briggs DR, Croxall JP, Naito Y, Kato A. 1992.** Diving pattern and performance in relation to foraging ecology in the gentoo penguin, Pygoscelis papua. *Journal of Zoology* **227**:211–230 DOI 10.1111/j.1469-7998.1992.tb04818.x.

**Wilson K, Lance M, Jeffries S, Acevedo-Gutiérrez A. 2014.** Fine-scale variability in harbor seal foraging behavior. *PLOS ONE* **9**:e94208 DOI 10.1371/journal.pone.0092838.

**Wilson RP, Putz K, Bost CA, Culik BM, Bannasch R, Reins T, Adelung D. 1993.** Diel dive depth in penguins in relation to diel vertical migration of prey: whose dinner by candlelight? *Marine Ecology Progress Series* **94**:101–104 DOI 10.3354/meps094101.

**Wilson RP, Pütz K, Peters G, Culik B, Scolaro JA, Charrassin JB, Ropert-Coudert Y. 1997.** Long-term attachment of transmitting and recording devices to penguins and other seabirds. *The Wildlife Society Bulletin* **25**:101–106.

**Wing SR. 2009.** Decadal-scale dynamics of sea urchin population networks in Fiordland, New Zealand are driven by juxtaposition of larval transport against benthic productivity gradients. *Marine Ecology Progress Series* **378**:125–134 DOI 10.3354/meps07878.

**Wing SR, Jack L. 2013.** Marine reserve networks conserve biodiversity by stabilizing communities and maintaining food web structure. *Ecosphere* **4**:1–14 DOI 10.1890/es13-00257.1.

**Wing SR, Jack L. 2014.** Fiordland: the ecological basis for ecosystem management. *New Zealand Journal of Marine and Freshwater Research* **48**:577–593 DOI 10.1080/00288330.2014.897636.

**Ydenberg RC, Clark CW. 1989.** Aerobiosis and anaerobiosis during diving by western grebes: an optimal foraging approach. *Journal of Theoretical Biology* **139**:437–449 DOI 10.1016/s0022-5193(89)80064-5.

**Zamon JE, Welch DW. 2005.** Rapid shift in zooplankton community composition on the northeast Pacific shelf during the 1998 1999 El Niño La Niña event. *Journal of Theoretical Biology* **62**:133–144 DOI 10.1139/f04-171.