# Peer review of "Inter-colony and inter-annual behavioural plasticity in the foraging strategies of a fjord-dwelling penguin—good news in the face of environmental change?"

_PeerJ, doi:10.7717/peerj.19650_

## Round 0.1 · original submission · Minor Revisions

Thank you very much for your manuscript titled “Inter-colony and inter-annual behavioural plasticity in the foraging strategies of a fjord-dwelling penguin – good news in the face of environmental change?” that you sent to PeerJ.

This study presents very valuable and relevant information on the marine ecology of the study species and its impact on their feeding strategies, comparing two colonies over two years of study. The results showed that the two colonies displayed markedly different feeding preferences.

As you will see below, comments from both referees suggest a minor revision. Given this, I would like to see a minor revision dealing with the comments. I will be happy to accept your article pending further revisions, detailed by the referees.

Reviewer 1 suggests clarifying the concept of climate change and its interpretation within the discussion, as well as some details to be defined within the methods.

Reviewer 2 requests clarification regarding the experimental design and the inclusion of bibliographic support for the hypothesis. She also has some specific questions about specific phrases in the text.

Please note that we consider these revisions to be important and your revised manuscript will likely need to be revised again.

·

Basic reporting

The manuscript is clearly written with appropriate scientific tone and terminology. Still, I think there are something to be improved, especially in the discussion for interpreting the results.The background is well referenced in relation to penguin foraging and environmental influences such as ENSO and La Niña/El Niño effects. The manuscript are logically organized but some parts seem to be skipped with no enough explanation.
Figures are relevant and clearly presented, with informative captions. Tables are appropriately formatted and provide detailed statistical information.
The authors mention use of raw data and GPS tracks but hard to find where the data is deposited (e.g., repository). This should be clarified to comply with PeerJ's open data policy.

Experimental design

The study is within the scope of PeerJ and provides valuable insight into intra-specific behavioral variation in seabirds. The research question is well defined, focusing on behavioral plasticity in tawaki foraging strategies across colonies and years. The deployment of GPS-dive loggers and statistical analysis using GLMMs and LMMs are appropriate.
Ethical approval from the University of Otago and the Department of Conservation permit are noted and the manuscript specifies the necessary permits were obtained, fulfilling PeerJ’s requirements.

Validity of the findings

The datasets across two years, although modest in sample size, are reasonable considering the logistical challenges. The discussion is grounded in the results and literature, highlighting the unexpected flexibility of inner-fjord birds compared to outer-fjord birds. The manuscript acknowledges limitations related to the study's geographical scope (focus on one fjord) and suggests further research in other fjord systems. It should be further clearer throughout the manuscript, including Abstract and Discussion. Conclusions are well supported by the data, particularly regarding plasticity and conservation implications.

Additional comments

I think this manuscript contributes new and important data on the foraging plasticity of a lesser-studied penguin species. I clearly understand that it is highly limited to approach this bird. Still, this study stands on the two year data and the small sample size. There seem to be many other factors to affect their foraging trips. The authors explain linkages between environmental variability (ENSO/La Niña) and seabird foraging ecology, but I strongly suggest that it is toned down and carefully interpreted relation to climate change.
My major concerns are as below.
1. The manuscript often mixes the two terms, 'climate change' and 'environmental change'. As I understand, climate change is dealing with the long-term climate effects while environmental change is a broader term including climate change. It can be clearer in the introduction and discussion.
2. In the methods, it seems that accelerometers were equipped, but it is not mentioned in the results and discussion. It should be mentioned why it is omitted in the analysis.
3. In the discussion, the authors concluded that this plasticity may help penguins be advantageous with the climate change. Based on the two year data and the small sample size, I am concerned about the interpretation. I would rather focus on the behavioral plasticity and other possibilities.
4. As I undertand, La Niña conditions in New Zealand often regarded to make it harder for penguins to breed and feed, which can affect the negative impacts on chick survival. What do authors think about this result that La Niña may favorable for only this species, unlike other mainland birds? Does this bird have a different strategy?
5. Can La Niña conditions delay breeding later in 2020 like the case in blue penguins (Perriman et al. 2000, New Zealand Journal of Zoology)? How about the prey distribution or composition changes with La Niña?
6. Dive data seem to be filtered at 0.5m depth and dive seconds duration. Is there a reference for that? Please provide a supporting reference. I am concerned if it may overestimate the number of dives.

Reviewer 2 ·

Basic reporting

No comment

Experimental design

Please explain why only guard-stage females were chosen for the study. Was this to control for potential sex-driven differences? The authors state that they determined sex based on weight and beak measurements, but did not include a citation for this method or relevant study. Is this method specific to the tawaki?

Validity of the findings

No comment

Additional comments

I really like the hypothesis that chick-rearing tawaki head towards the open ocean until they encounter adequate foraging conditions. Is there any literature out there on this to further support it? Perhaps not necessarily in seabirds?

Minor additional comments:

1. Line 241: Please explain how you defined a "complete trip"

2. Line 352: First sentence needs rewording. Example “Tawaki exhibited colony-level differences in diving and foraging patterns. Specifically, differences between colonies located inside the fjord vs the entrance were more pronounced…”

3. Line 420-423: The foraging tracks suggest tawaki can travel large differences (at least 40km), so I am not sure if this a valid argument.

4. Line 449: This sentence needs rewording. “Appeared” suggests that it was observed that prey density was plentiful. Perhaps say “fjord prey for Harrison Cove birds was likely plentiful…”

5. In figure 2, why are some of the yellow Harrison cove tracks over land?

---

## Round 0.2 · accepted · Accept

After reviewing this revised version of your manuscript, I see that the main comments suggested by the reviewers have been included, while the suggestions not considered are justified in detail. Therefore, I am satisfied with the current version and consider it ready for publication.